

# MtDNA species-level phylogeny and delimitation support significantly underestimated diversity and endemism in the largest Neotropical cichlid genus (Cichlidae: *Crenicichla*)

Oldřich Říčan[1], Klára Dragová[1], Adriana Almirón[2], Jorge Casciotta[3,4], Jens Gottwald[5] and Lubomír Piálek[1]

[1] Faculty of Science, Department of Zoology, University of South Bohemia, České Budějovice, Czech Republic
[2] División Zoología Vertebrados, UNLP, Facultad de Ciencias Naturales y Museo, La Plata, Buenos Aires Province, Argentina
[3] CIC, Comisión de Investigaciones Científicas de la Provincia de Buenos Aires, Buenos Aires, Argentina
[4] UNLP, Facultad de Ciencias Naturales y Museo, División Zoología Vertebrados, Paseo del Bosque, La Plata, Buenos Aires Province, Argentina
[5] Heinrich-Lödding-Str. 14, 30823 Garbsen, Garbsen, Germany

Corresponding author
Oldřich Říčan,
oldrich.rican@prf.jcu.cz

## ABSTRACT

*Crenicichla* is the largest and most widely distributed genus of Neotropical cichlids. Here, we analyze a mtDNA dataset comprising 681 specimens (including *Teleocichla*, a putative ingroup of *Crenicichla*) and 77 out of 105 presently recognized valid species (plus 10 out of 36 nominal synonyms plus over 50 putatively new species) from 129 locations in 31 major river drainages throughout the whole distribution of the genus in South America. Based on these data we make an inventory of diversity and highlight taxa and biogeographic areas worthy of further sampling effort and conservation protection. Using three methods of molecular species delimitation, we find between 126 and 168 species-like clusters, *i.e.*, an average increase of species diversity of 65–121% with a range of increase between species groups. The increase ranges from 0% in the Missioneira and Macrophthama groups, through 25–40% (Lacustris group), 50–87% (Reticulata group, *Teleocichla*), 68–168% (Saxatilis group), 125–200% (Wallacii group), and 158–241% in the Lugubris group. We found a high degree of congruence between clusters derived from the three used methods of species delimitation. Overall, our results recognize substantially underestimated diversity in *Crenicichla* including *Teleocichla*. Most of the newly delimited putative species are from the Amazon-Orinoco-Guiana (AOG) core area (Greater Amazonia) of the Neotropical region, especially from the Brazilian and Guiana shield areas of which the former is under the largest threat and largest degree of environmental degradation of all the Amazon.

## INTRODUCTION

The cichlid family (Cichlidae) is an important family of Neotropical fishes that is the dominant group of larger sized fishes in Middle America (*Myers, 1966*; *Bussing, 1976*, *1985*; *Říčan et al., 2013*, *2016*) and the third richest family of fishes in South America (*Van der Sleen & Albert, 2018*). *Crenicichla* Heckel, 1840 is the most widely distributed and largest cichlid genus in South America and in the whole Neotropical biogeographic region (*Ploeg, 1991*; *Piálek et al., 2012*; *Van der Sleen & Albert, 2018*) and is the eighth richest fish genus in South America (*Van der Sleen & Albert, 2018*). The Neotropical cichlids are members of the by far most diverse and extreme aquatic ecosystems on Earth and the largest community of freshwater fishes in the world with more than 5,600 valid species and with current estimates far exceeding 7,000 species, representing about 10% all living vertebrate species (*Reis, Kullander & Ferraris, 2003*; *Albert & Reis, 2011*; *Van der Sleen & Albert, 2018*).

*Crenicichla* species inhabit a wide range of virtually all cis-Andean (east of the Andes) South American water bodies (*Stawikowski & Werner, 2004*; *Kullander, López-Fernández & van der Sleen, 2018*) from small rainforest and mountain streams, medium through large rivers, lakes, floodplains and even marshes, living both in main channels as well as pools and riffles and even rapids, but generally few species are widely distributed or found in more than one type of habitat, with endemism, regionality, and habitat adaptations being strongly pronounced in this genus as well as in cichlids in general. *Crenicichla* are monogamous substratum spawning biparental fishes with a strong nest and fry protection and very long parental care (*Stawikowski & Werner, 2004*), even among the generally long parental care in cichlids in general (*Stawikowski & Werner, 2004*; *Kullander, López-Fernández & van der Sleen, 2018*). Its representatives cover a wide range of body sizes from small (though not very small) to some of the largest Neotropical cichlids and are predominantly and also ancestrally specialized and often among the top predators, especially in smaller sized rivers and streams. The smaller sized species feed mainly on benthic (but also terrestrial) invertebrates and small fishes, while the larger and large species are predators of fishes and large invertebrates (*Stawikowski & Werner, 2004*; *Kullander, López-Fernández & van der Sleen, 2018*).

*Crenicichla* species diversity is centered on the Amazon-Orinoco-Guiana (AOG) core area of the Neotropical region (Greater Amazonia) (*Stawikowski & Werner, 2004*), which is also the center of diversity for virtually all Neotropical fish groups (*Albert & Reis, 2011*; *Van der Sleen & Albert, 2018*). Most species groups of the genus co-occur in this core (*Ploeg, 1991*; *Stawikowski & Werner, 2004*; *Piálek et al., 2012*), but the most morphologically and ecologically distinct species and the only species that depart from their ancestral predatory ecomorphology are found in the southern subtropical region of the Neotropics, more specifically only in the La Plata river basin, and here only in the Middle Paraná and its major affluent, the Iguazú river, and in the Uruguay river (*Piálek et al., 2012*; *Burress et al., 2018a*, *2018b*). These are the only *Crenicichla* assemblages where we find specialized molluscivorous species, specialized large-lipped invertebratophagous crevice feeders, and even partially herbivorous species (periphyton

grazers), together with both benthic as well as pelagic invertebratophagous-piscivorous predators, and all or some of these ecomorphs occur in sympatry in the Middle Paraná and the Uruguay river basins and have most probably evolved in sympatry (*Piálek et al., 2012*, *2019a*, *2019b*; *Burress et al., 2018a*, *2018b*).

*Ploeg (1991)*, while quite outdated, is still the main reviewer of *Crenicichla* systematics and the author of most of the informal species groups within the genus. The species groups differ in many parameters including body size, head and body proportions, coloration patterns as well as in ecological characteristics and preferred habitats, and for some groups in biogeography. Apart from *Ploeg (1991)*, initial species group divisions were also introduced by *Kullander (1991*, *1997)* and the most comprehensive review of all aspects of *Crenicichla* diversity is given by *Stawikowski & Werner (2004)*. The following species groups within the genus have thus been proposed so far: Acutirostris, Lacustris, Lugubris, Macrophthalma, Missioneira, Reticulata, Saxatilis, Scottii, and Wallacii. Based on presently valid numbers of species the largest species groups are Saxatilis and Lacustris, both with 25 species. The distributions of the Acutirostris, Lugubris, Macrophthalma, Reticulata, Saxatilis, and Wallacii groups are centered on the AOG core of South America where they all overlap in their distributions but generally prefering different habitats, while the Lacustris, Missioneira, and Scottii groups are endemic to SE South America (chiefly the SE Atlantic coast, the La Plata river basin and the northern Pampean region of Argentina) outside of the AOG core. The Lacustris and Missioneira groups include the sympatric species flocks of highly ecomorphologically distinct species (*Piálek et al., 2012*, *2019a*, *2019b*; *Burress et al., 2018a*, *2018b*), while the AOG core centered species groups are internally much more ecomorphologically homogeneous.

The type species of the genus is *Crenicichla macrophthalma* Heckel, 1840, the sole representative of the Marcrophthalma species group, a morphologically rather recognizable species with large eyes. *Crenicichla* contains two genus-level names currently considered as synonyms, *Batrachops* Heckel, 1840 (*Kullander, 1986*) and *Boggiania* Perugia, 1897 (*Kullander, 1986*).

*Crenicichla* currently includes 132 nominal species, of which 96 are generally considered valid and 36 as synonyms (www.cichlidae.com), but this division has differed between authors quite widely (*Ploeg, 1991*; *Stawikowski & Werner, 2004*; *Kullander, López-Fernández & van der Sleen, 2018*). *Crenicichla* is thus currently the largest Neotropical cichlid genus, followed by *Apistogramma* (with 106 nominal species, 93 of which are generally considered valid and 13 synonyms; www.cichlidae.com). A list of valid species of *Crenicichla* with complete species names including author (plus a division of the valid species into the species groups), plus nominal species presently considered as synonyms are given in alphabetical list in Appendix 1, and by year of description in Appendix 2.

The real species diversity in *Crenicichla* is, however, based on preliminary and predominantly unpublished information (pers. obs.) or information published only in cichlid hobby literature (*Stawikowski & Werner, 2004*) quite probably significantly underestimated. At least 85 putative undersribed species have been proposed in the literature (summary in *Stawikowski & Werner, 2004*) (Appendix 3). The large number of proposed putatively undescribed species in the genus have predominantly been identified

thanks to the collecting efforts of the cichlid hobbyists. *Stawikowski & Werner (2004)* is the main review of systematics, literature and species diversity, provides the characterization and photographs of the majority of the nominal as well as the putative undescribed species, including their proposed classification into species groups.

In 1988 Kullander described a new cichlid genus *Teleocichla Kullander, 1988* with then six species of small elongated cichlids, the only cichlids in the Neotropics morphologically similar to *Crenicichla* (*Kullander, 1988*). The type species of that genus is *Teleocichla centrarchus Kullander, 1988* and the genus presently contains nine species (Appendices 1–3) all endemic to the Eastern Amazon within the AOG core of the Neotropics. *Teleocichla* species are superficially most similar to the Wallacii group of *Crenicichla*, mostly due to their similarly small size, but are distinct from the Wallacii group and from all *Crenicichla* in many morphological characters, especially in the very distinct morphology of the head (especially a short, downturned snout and short jaws with upper slightly projecting in front of lower *vs.* the contrary) and also body (reduced swim bladder, lateral lines and pelvic-fin shape) with obvious specializations for a benthic and rheophilic mode of life in rapids (*Kullander, 1988*; *Ploeg, 1991*). Parallel but distinct adaptions for a similar lifestyle are also found in some sympatric members of the Lugubris group of *Crenicichla*, which are however large piscivorous species unlike the small benthic-invertivorous *Teleocichla* species (*Kullander, 1988*; *Ploeg, 1991*; *Stawikowski & Werner, 2004*).

*Kullander (1988)* hypothesized *Teleocichla* to be closely related to *Crenicichla* (they are the only similar groups of cichlids in the Neotropics) but unequivocal synapomorphies are few (*Kullander, 1988*; *Ploeg, 1991*). The close relationship of these genera is straightforward, but a more intriguing question is whether *Teleocichla* is a separate genus from *Crenicichla*, especially given the large diversity within the latter. *Kullander (1988)* gave many characters that are different in the two genera, but they are either correlated with habitat (benthic and rheophilic in *Teleocichla vs.* extremely wide ranging in *Crenicichla*), variable, or unconvincing and not evidently synapomorphic for either genus (*Kullander, 1988*; *Ploeg, 1991*). Ever since the description of *Teleocichla* its separate generic status has thus been questioned (summary in *Stawikowski & Werner, 2004*) and *Ploeg (1991)* in the so far only formal review even synonymized *Teleocichla* with *Crenicichla*. The reasons for the lack of general acceptance of *Teleocichla* were its not completely convincing diagnosis, that it has not been included in the morphological phylogeny of *Kullander (1998)* or any subsequent morphological phylogeny, and molecular phylogenies, which are actually strongly suggesting that *Teleocichla* is one of the species groups of *Crenicichla* (*Piálek et al., 2012*; *Ilves, Torti & López-Fernández, 2018*; *Burress et al., 2018a*, the so far most thorough study with the most included species and the largest genetic dataset for both genera).

This study is designed as a first major step in a series of contributions that will review the species diversity of *Crenicichla* (including *Teleocichla*) because as outlined above it is (1) the largest Neotropical cichlid genus, (2) with the largest number of proposed putatively undescribed species and (3) because it is an important component of South American fish communities, where it plays an important ecological role and has the

potential to become a useful biogeographic and ecological bioindicator (due to its high endemism and narrow habitat preferences). A good knowledge of the species diversity of the genus is thus of key importance.

In this study we have assembled a mtDNA (ND2 and cytb) dataset of 681 ingroup specimens representing 77 out of 105 presently recognized valid species and 10 out of 36 nominal species treated as synonyms (Appendices 1 and 2) plus over 50 of the at least 85 postulated putative species (Appendix 3). With the use of phylogenetic analyses of this dataset including a dated one and together with three molecular species delimitation methods we test the species diversity hypothesis within the genus (*i.e.*, valid species, synonymized species and postulated putative species) that has been established based on traditional characters. This same specimen sampling with future additions will be in the consecutive studies used for a nuclear DNA phylogeny reconstruction (ddRAD and whole genome) to compare with the here presented results based on mtDNA and for morphological analyses and revisions.

## MATERIALS & METHODS

### Specimen sampling

This study is based on a total of 681 *Crenicichla* individuals (including *Teleocichla*) (Appendix 1, Table S1) that were collected throughout the whole distribution of the two genera in South America (Fig. 1) from 129 locations (La Plata 28, E. coast 10, W. Amazon 17, Orinoco 19, Negro 6, E. Guiana shield 24, E. Amazon 25) in 31 major river drainages (=ecoregions *sensu* Abell et al., 2008; La Plata 7, E. coast 6, W. Amazon 3, Orinoco 5, Negro 1, E. Guiana shield 3, E. Amazon 6).

Tissue samples for DNA analysis were taken from the right pelvic fin, stored in 95% ethanol and deposited in the tissue collection of the Department of Zoology, Faculty of Science, University of South Bohemia. Voucher specimens were photographed alive, permanently tagged and identified using a unique code (C1 to C704; Table S1), and following euthanasia (overdose of anaesthetic MS-222 Tricaine methanesulfonate and Benzocaine hydrochloride dissolved in water) preserved in 10% formalin and later transferred to 70% ethanol. Voucher specimens are deposited in the fish collection of the Department of Zoology, Faculty of Science, University of South Bohemia, and in the fish collections of Museo de La Plata (MLP), Museo Argentino de Ciencias Naturales (MACN), and Asociación Ictiológica La Plata (AI). Study of the animals was approved by the Control Commission for Ethical Treatment of Animals, University of South Bohemia, Faculty of Science, Czech Republic (17864/2005-30/300).

Specimens for the study were obtained by extensive collecting lasting several decades through our collecting and a network of dedicated cichlid hobbyists, cichlid fish importers, and colleagues who have collected and kept animals over this period. Our collecting was done under permanent collection permits to Hernán Ortega (National University of San Marcos, Lima, Peru), Javier A. Maldonado Ocampo (Pontificia Universidad Javeriana, Bogotá, Colombia), Cecilia Rodríguez-Haro (Universidad Regional Amazónica IKIAM, Tena, Ecuador), and Nadacion FUDECI (Venezuela) under bilateral agreements, and under temporary permits from the Administración de Parques Nacionales y Ministerio

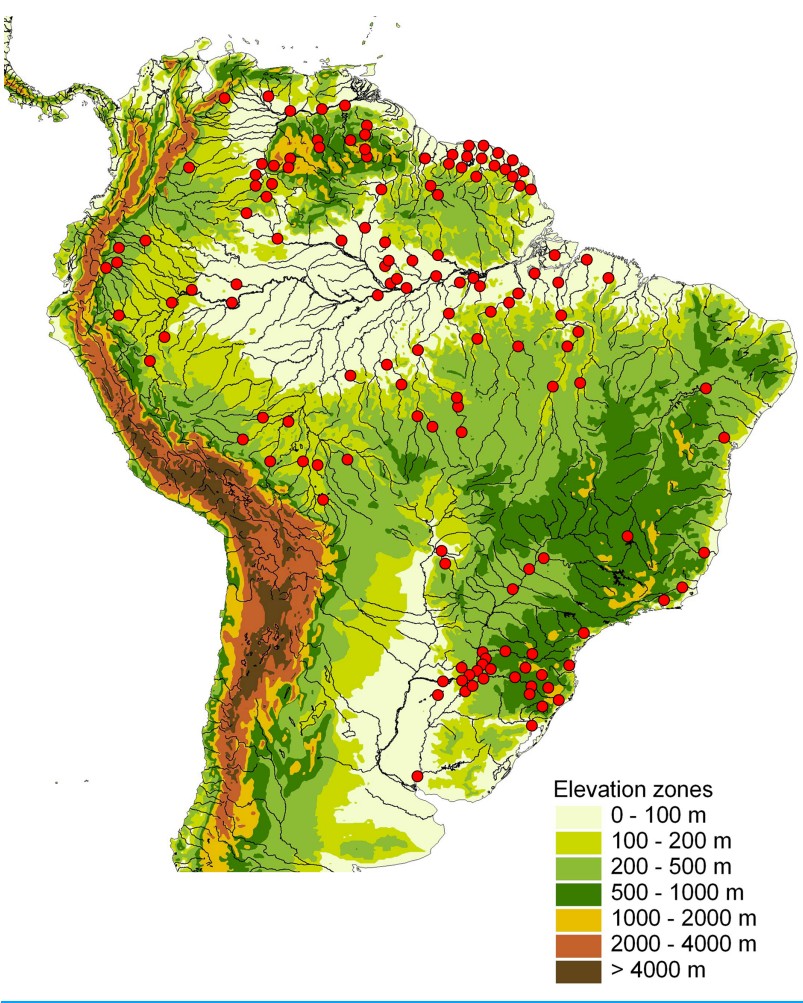

**Figure 1 Map of sampling locations shown by red dots on a relief map of South America.** One dot may represent several localities in close proximity.

de Ecología y Recursos Naturales Renovables, Argentina (permits nos. NEA328 rnv3 and Res: 509/07, respectively). Specimens from Brazil and the Guianas were collected and imported by a consortium of professional aquarium fish importers coordinated by JG (Amazon Peixes Ornamentais LTDA, Manaus, Brazil; Netuno Aquarium Peixes Orn LTDA, Manaus, Brazil; Amazonstar Comercio de Peixes Ornamentais LTDA, Manaus, Brazil; FLOCAMBUS S.A.R.L, French Guiana; Oleeja Ornamental Fish Export, Suriname, South America; and Panta Rhei GmbH, Hannover, Germany) (permits nos. 0121/ UVAGRO/AERO/16, 0545/UVAGRO/AERO/15, 00101/SVA-AIRJ/LPS L, 05021/ SVAGRU/10, 0229 UVAGRO-ABEL 15, 0938 UVAGRO-AEROBEL 14, 0462 UVAGRO-AEROBEL 15, 0069 UVAGRO-AEROBEL 20, 011013/2016/SVAGRU, 004989/2020/VIGIAGRO, 00006331/2020-SVA-GRU, 1351 SVA-GRU 13, 2012403102A, and 201804004E). The imported specimens are kept alive and are being reared in care of JG, in a network of fish hobbyists coordinated by JG and voucher specimens are kept at the fish collection of the Department of Zoology, Faculty of Science, University of South Bohemia. Collection localities are given in Table S1 and are visualized in Fig. 1.

## Species determination

We combine morphological species determination with post-hoc species delimitation using molecular mtDNA cytochrome b (*cytb*) and ND2 markers following the protocol of *Říčan et al. (2019)*. Specimens were identified to species with the use of original descriptions, identification keys, and comparative material. Specimens that are new undescribed species or that could not be identified to species level are reported as "*Crenicichla* sp." (possibly new/unidentified species), specimens that were identified to species only with reservation are reported as "*Crenicichla* cf. species" (a species that conforms to diagnosis with reservation and occurs outside its supposed distribution, possibly new) or "*Crenicichla* aff. species" (closely related species, revealed by DNA phylogeny, possibly new). For summary statistics, "*C.* aff. species" and "*C.* cf. species" are considered equivalent to "*C.* species".

## Molecular markers

For molecular phylogenetic analyses and delimitation of putative species we have used two mitochondrial (mtDNA) markers, the cytochrome *b* (*cytb*) gene and the ND2 gene following the protocol of *Říčan et al. (2019)*. The *cytb* is the single most often used molecular marker in cichlid phylogenetic studies and has been one of the first molecular markers to provide well resolved phylogenies of the Neotropical cichlids that have subsequently been virtually completely confirmed by later studies including recent phylogenomic studies (*e.g. Říčan et al., 2016*; *Ilves, Torti & López-Fernández, 2018*; *Burress et al., 2018a, 2018b*). The ND2 gene has been found to have in *Crenicichla* even better and more evenly distributed support values across both shallow and deep nodes in the phylogeny than the cytb (*Piálek et al., 2012*).

## Laboratory methods

Genomic DNA was extracted from ethanol-preserved fin tissue using the JETQUICK Tissue DNA Spin Kit (Genomed, Hannover, Germany) following standard protocol and as described in *Říčan et al. (2019)*. The primers and reaction conditions of polymerase chain reaction (PCR) amplification are as in *Říčan, Zardoya & Doadrio (2008)*. The products were analyzed in an ABI 3730XL automated sequencer (Applied Biosystems; Macrogen Inc., Seoul, Korea). Contiguous sequences of the gene segments were created by assembling DNA strands (forward and reverse) using GENEIOUS v. 11.0.2 (http://geneious.com, *Kearse et al., 2012*). Nucleotide coding sequences were also translated into protein sequences to check for possible stop codons or other ambiguities. All newly generated sequences (558) were deposited in GenBank under Accession numbers MW554711–MW584412 (Table S1). Sequences were aligned using MUSCLE v. 3.8 (*Edgar, 2004*), using the default settings.

## Phylogenetic methods

As outgroups we have included *Astronotus ocellatus* and *Satanoperca jurupari* based on previous knowledge of cichlid relationships and studies dedicated to *Crenicichla-Teleocichla* and its most closely related groups (*Piálek et al., 2012*;
*Ilves, Torti & López-Fernández, 2018*; *Burress et al., 2018a*). Maximum parsimony (MP) analysis in PAUP* 4b.10 (*Swofford, 2003*), maximum likelihood (ML) analyses in RAxML v8.2.4 (*Stamatakis, 2014*) and Bayesian inference analyses (BI) in MrBayes v3.1.2 (*Huelsenbeck & Ronquist, 2001*; *Ronquist & Huelsenbeck, 2003*) and BEAST v1.8.4 (*Drummond & Rambaut, 2007*) were used for phylogenetic inference following the protocol of *Říčan et al. (2019)*. The MP phylogenetic analyses in PAUP* were run with 500 random sequence additions, 10 trees kept per addition, and a hs (heuristic) search on the saved trees to find all the shortest trees. Bootstrap analyses were done using the same approach, with five random sequence additions per one replication. Bootstrap analyses were run with 1,000 replications. The BI analyses in MrBayes (and BEAST, see below) were run with partitioning into codon positions (1st + 2nd *vs*. 3rd). An optimal model of evolution according to Akaike criterion was selected using MrModeltest 2.2 (*Nylander, 2004*) and PAUP* v. 4.0b10 (*Swofford, 2003*). The BI analysis using the Markov chain Monte Carlo (MCMC) simulation was run for two million generations with trees sampled and saved every 1,000 generations. Two independent analyses, each comprising two runs with eight chains, were performed to compare results of independent analyses. The analyses were run at the freely available Cipres server (https://www.phylo.org/) and in the Czech academic National Grid Infrastructure MetaCentrum (www.metacentrum. cz). The first 10% of trees from each run were discarded as burn-in. Convergence of the runs was estimated with the use of graphical visualization and diagnostics (especially the effective sample size; ESS) in Tracer v. 1.8.4 (*Rambaut et al., 2018*). The remaining trees were used for reconstruction of the 50% majority-rule consensus tree with posterior probability (PP) values of the branches.

## Molecular clock dating analyses in BEAST

For divergence time estimation we used the Bayesian Evolutionary Analysis by Sampling Trees (BEAST) software package version v.1.8.4 (*Drummond & Rambaut, 2007*) with parameters as in MrBayes analyses (except for 30 million generations). We used the relaxed molecular clock model with lognormal distribution of rates and for tree prior the coalescent model with constant size. The calibration of the molecular clock was performed using secondary calibration from the study of *Musilová et al. (2015)* which was focused on the dating of Neotropical cichlids. *Musilová et al. (2015)* employed a calibration using a set of Neotropical fossil cichlid species and the study is so far the best sampled dated phylogeny of the Neotropical cichlids. For the calibration of *Crenicichla* we have used the basal node of the genus, estimated by *Musilová et al. (2015)* at 28 Ma (normal distribution; SD = 1).

The analyses were run at the freely available Cipres server (https://www.phylo.org/). Runs were checked for convergence with Tracer v.1.8.4 (*Rambaut et al., 2018*). Four well converged runs were combined in LogCombiner v.1.8.4 with a burnin of 10% for each of the data partition schemes. The final tree for each data partition scheme was produced from these data with TreeAnnotator v.1.8.4.

## Species delimitation analyses using GMYC and PTP

We employed the General Mixed Yule Coalescent (GMYC) and Poisson tree processes (PTP) analyses for molecular species delimitation using the studied markers and following the protocol of *Říčan et al. (2019)*. Both methods were designed for delimiting species based primarily on single molecular markers (hence where multilocus coalescent-based methods are not applicable).

The General Mixed Yule Coalescent (GMYC) model (*Pons et al., 2006*; *Fujisawa & Barraclough, 2013*) is frequently used in empirical studies (*Fontaneto et al., 2007*; *Monaghan et al., 2009*; *Carstens & Dewey, 2010*; *Vuataz et al., 2011*; *Powell, 2012*) and the newer Poisson tree processes (PTP) model (*Zhang et al., 2013*) has been shown to even outperform the GYMC method where distances between species are small. Both methods outperform OTU-picking methods (relying on simple sequence similarity thresholds) and are more robust to cases where the barcoding gap is absent (*Zhang et al., 2013*). Both methods model speciation (among-species branching events) *via* a pure birth process and within-species branching-events as neutral coalescent processes. The methods identify the transition points between inter- and intra-species branching rates by maximizing the likelihood score of the model. While the GMYC method uses time to identify branching rate transition points (hence only on a time-calibrated ultrametric tree) the PTP method directly uses the number of substitutions and does not require a time-calibrated ultrametric tree. Both methods assume that all lineages leading from the root to the transition points are different species.

PTP and GMYC analyses were run at the freely available web interface (http://species.h-its.org and https://mptp.h-its.org). Prior to analyses a haplotype dataset was created in Fabox (*Villesen, 2007*). The mPTP and bPTP analyses were run on the MrBayes haplotype tree and the GMYC analysis on the ultrametric haplotype BEAST tree.

## RESULTS

This study is based on the by far largest sampling of *Crenicichla* specimens (including *Teleocichla*) and species. The aligned mtDNA matrix comprised 681 specimens plus outgroups with 2,163 aligned base pairs of the concatenated cytb and ND2 markers. The ingroup dataset comprised a total of 591 unique haplotypes. Phylogenetic hypotheses generated from this dataset using MP, ML, and BI analyses are highly congruent and the majority of the nodes within species groups and at the level of species (the focus of this study) are well supported (PP > 95%, BS > 75%; cf. Fig. 2 and Figs. S1, S2A and S2B) except for parts of the Missioneira and Saxatilis groups as detailed below. The dated BI BEAST topology (Figs. 2–6) is used throughout the text to describe and discuss results unless otherwise noted.

*Teleocichla* is in the mtDNA BI BEAST phylogeny the sister-group of the genus *Crenicichla*, while in RAXML phylogeny it is nested within *Crenicichla*, but above two of the unsupported nodes. The following supragroup relationships are the same in the BI BEAST (Figs. 2–6) and RAXML (Figs. S1, S2A and S2B) phylogenies. One clade includes the Missioneira group plus Lacustris (the Scottii group is firmly nested within the latter; Figs. 2 and 3–6), followed by the Macrophthalma, followed by the Reticulata.

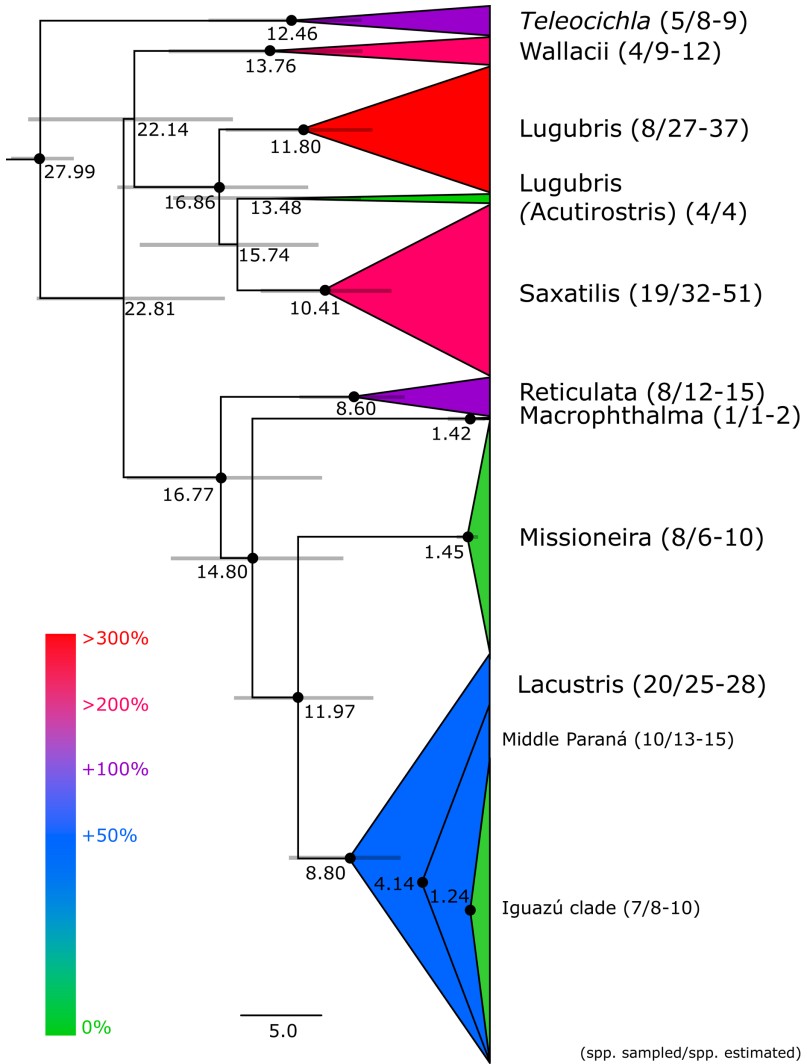

**Figure 2 Maximum clade credibility chronogram from 10,000 posterior trees generated using Beast collapsed to show species groups (same tree as in Figs. 3–6).** Dataset comprised of 591 unique haplotypes (from a total of 681 samples) of *Crenicichla* (including *Teleocichla*) cytb and ND2 sequences (2,163 aligned base pairs). Bayesian posterior probabilities above 0.95 are shown as black points on nodes. Numbers at nodes show ages of divergence for selected nodes. Grey bars at nodes show 95% HPD intervals for reconstructed ages of nodes. Numbers behind species groups are sampled valid species/delimited species (range between methods based on Figs. 3–6). Colors of species groups show percentual increase of delimited species (average; based on color scale to the left) compared to presently valid number of species in a given species group.  

The remaining species groups form a second clade in the BEAST analysis (Fig. 2) where the Saxatilis is the sister group to the Acutirostris, followed by the Lugubris, followed by the Wallacii (Figs. 2 and 3–6). In the RAXML analysis (Figs. S1, S2A, S2B) the Saxatilis is the sister group to the Lugubris, followed by the Acutirostris, and the Wallacii is the sister group to the remaining groups including *Teleocichla*. The conflicting supragroup positions between the BEAST and RAXML analyses are without statistical support in either of the analyses.

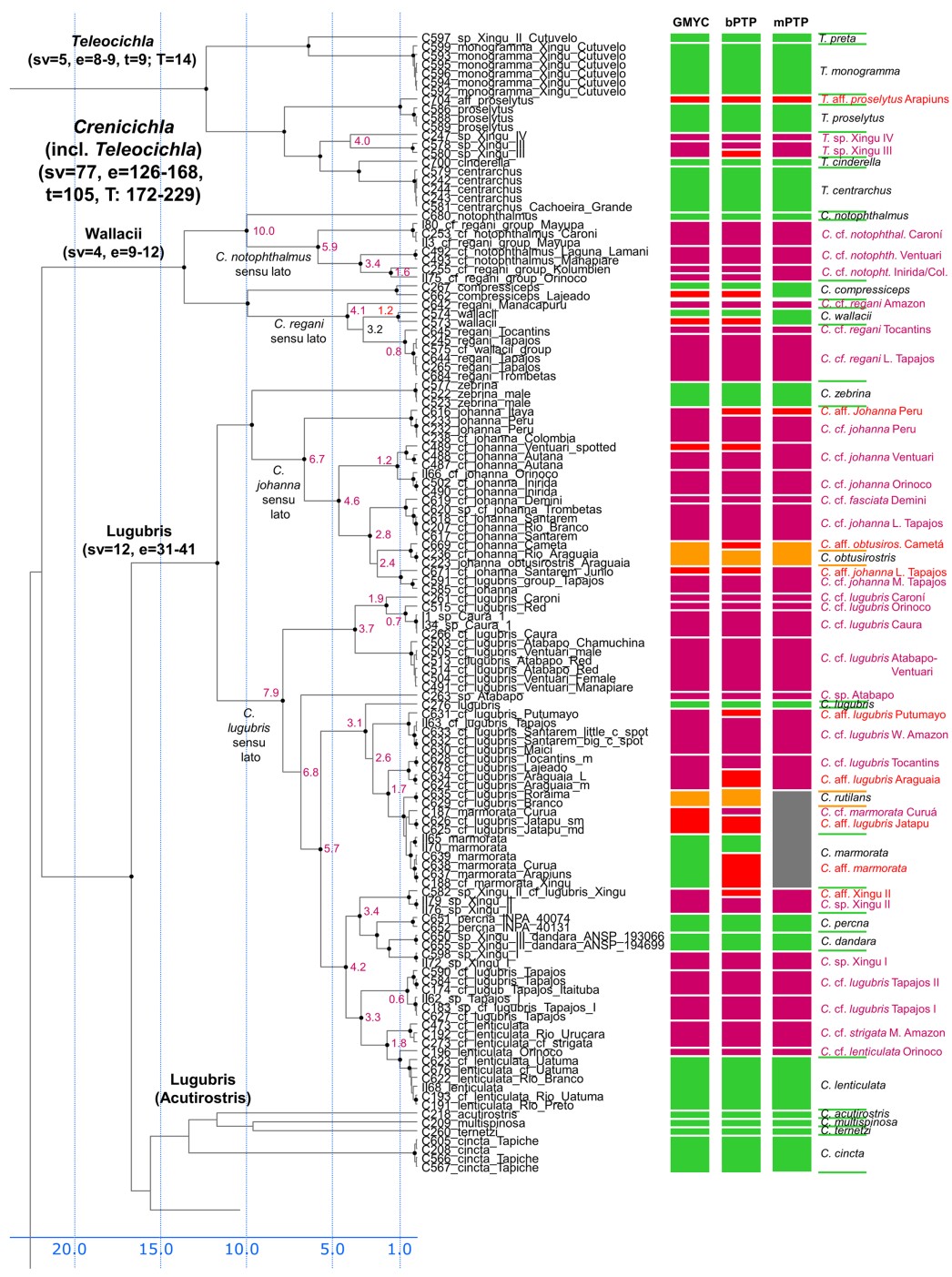

**Figure 3 First part of the maximum clade credibility chronogram from 10,000 posterior trees generated using Beast (same tree as Fig. 2) showing the Wallacii and Lugubris species groups and *Telecichla*.** Bayesian posterior probabilities above 0.95 are shown as black points on nodes. Numbers at nodes show ages of divergence for selected nodes. Point estimate species delimitations are shown by method as coloured boxes. Colour of the box shows correspondence to presently valid classification: Green: match with a presently recognized valid species; Orange: delimited taxon is currently treated as a synonym of a valid species; Violet: delimited taxon has been proposed in literature as a putative new species; Red: newly delimited taxon not previously suggested in literature; Grey: conservative delimitation

**Figure 3** (continued)
compared to valid classification (*i.e.*, contains more than one valid species). Abbreviations: sv, number of sampled valid species; e, estimated species based on the delimitations; t, total known valid species; T, extrapolated total species based on ratio of included valid species in analysis and all valid species known. mPTP analysis delimited 126 species, GMYC 153 species and bPTP 168 species in the whole species tree.                                                           

Description of the detailed species-level relationships is beyond the scope of the present paper but are well depicted in Figs. 3–6, in Table S1, and the overal species-level diversity patterns are described in a concise form below and in more detail in the Discussion. The overal estimated number of species derived from delimitation analyses ranged from 126 putative species (mPTP), through 153 (GMYC) to 168 (bPTP) (Table S1; Figs. 2–6). The total number of species delimited in *Crenicichla* (including *Teleocichla*) in this study is thus at least twice as high (126–168) as the number of included valid species (77).

The presently recognized valid species are mostly found well supported in our analyses. Of the 77 included valid species our analyses supported all but several weakly diagnosed species in the Saxatilis group (*C. proteus/C. lucius*, *C. inpa/C. alta*, *C. nickeriensis*, *C. menezesi*, *C. coppenamensis*, *C. albopunctata*), and then the rapidly diverged but morphologically distinct species of the Missioneira group (*C. missioneira*, *C. minuano*, *C. tendybaguassu*, *C. hadrostigma*, *C. celidochilus*) which are not found monophyletic even under our dense sampling. Of the 77 included valid species all three delimitation analyses supported 62 species, and at least two supported 64 species.

All 10 (out of 36) included taxa presently treated as synonyms are also all supported by most analyses as distinct from the species with which they are currently synonymized (shown in orange in Figs. 3–6; Appendices 1–3, Table S1; *Crenicichla biocellata* von Ihering, 1914; *Crenicichla cardiostigma Ploeg, 1991*; *Crenicichla dorsocellata* Haseman, 1911; *Crenicichla elegans* Steindachner, 1881; *Crenicichla guentheri Ploeg, 1991*; *Crenicichla menezesi Ploeg, 1991*; *Crenicichla obtusirostris* Günther, 1862; *Crenicichla punctulata* (Regan, 1905); *Crenicichla rutilans* (Jardine, 1843)). *Crenicichla edithae Ploeg, 1991* (Middle Rio Paraná) is currently treated as a synonym of *C. lepidota* (Rio Guaporé). We have not been able to include material from the type locality of *C. lepidota*, but other southern Amazonian cf. *lepidota* (Xingu, Tapajos, Teles Pires, Guapimirim) are distinct from *C. edithae* in our analyses and *C. edithae* additionally has moderate geographic structuring.

The described species based on our analyses form only half of the delimited species diversity in *Crenicichla*. The second half of the delimited *Crenicichla* species diversity is to a large extent formed by species that have been postulated as separate species in the cichlid hobby literature but have never been formally described. There are few of these species among the southern groups of *Crenicichla* (Missioneira group, Lacustris group) which have recently been revised (see Introduction), but they are prevalent among the Amazonian *Crenicichla* groups and in the Lugubris, Saxatilis and also Wallacii groups they actually form the majority of the delimited species diversity (shown in violet in Figs. 3–6; delimited species 31–41, 32–51, 9–12 *vs.* included valid species 12, 19, 4,

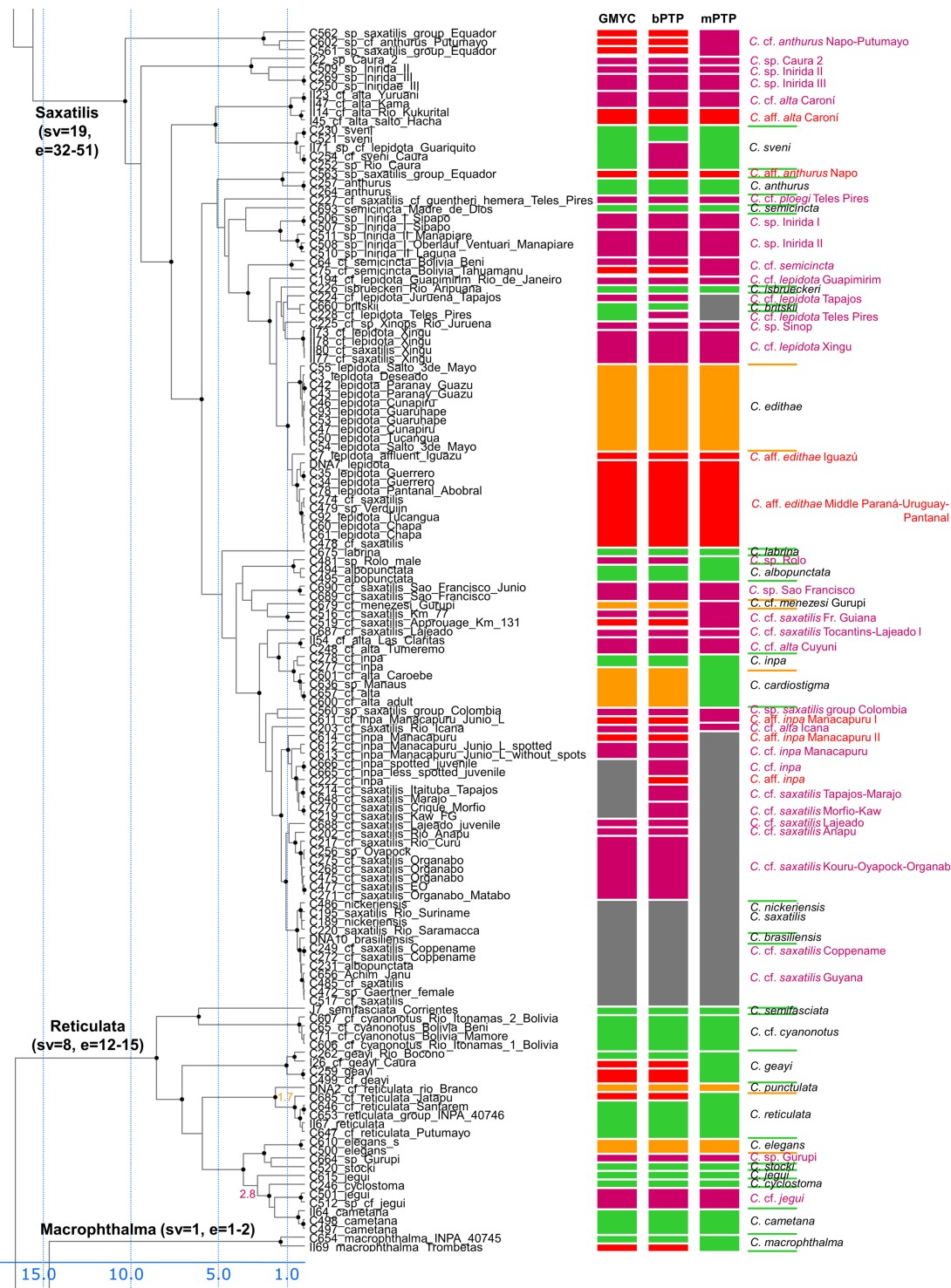

**Figure 4 Second part of the maximum clade credibility chronogram from 10,000 posterior trees generated using Beast (same tree as Fig. 2) showing the Saxatilis, Reticulata and Macrophthalma species groups.**

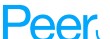

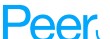

**Figure 5** **Third part of the maximum clade credibility chronogram from 10,000 posterior trees generated using Beast (same tree as Fig. 2) showing the Lacustris species group.**

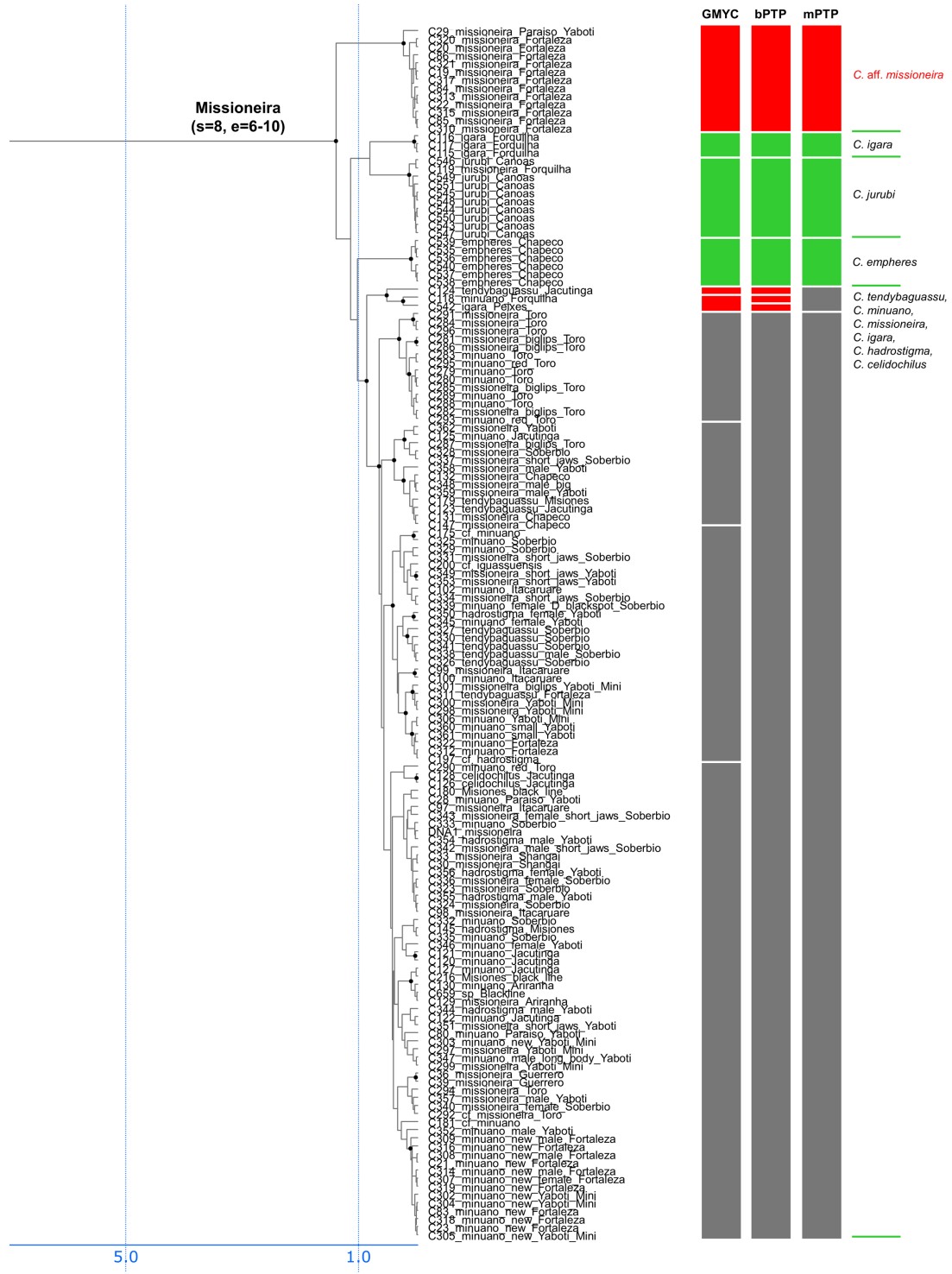

**Figure 6 Fourth part of the maximum clade credibility chronogram from 10,000 posterior trees generated using Beast (same tree as Fig. 2) showing the Missioneira species group.**

respectively). The most conservative estimate of these species is 40, given in the mPTP delimitation, while the GMYC and bPTP delimit 46 of them in common.

Finally, the remaining portion of the delimited putative species are those without any previous reference. These total a minimum of 18 delimited putative species in the mPTP analysis while 37 are the shared minimum between the GMYC and bPTP delimitations, and most analyses agree on 25 of these putative species, most of them again in the Lugubris group.

## DISCUSSION

### Phylogenetic relationships

Prior to this study the phylogenies with the most representative sampling of *Crenicichla* and *Teleocichla* were the studies of *Piálek et al. (2012)* and *Burress et al. (2018a)*. *Piálek et al. (2012)* included 161 specimens and 35 species in a multilocus mtDNA + nDNA phylogeny, while *Burress et al. (2018a)* included 57 species in a reduced genome representation nDNA ddRAD phylogeny. At the species level the relationships in *Piálek et al. (2012)* and *Burress et al. (2018a)*, derived from different molecular markers, are highly congruent and are also highly congruent with our present study with a much denser sampling, the only exceptions are within the Missioneira group and parts of the Saxatilis group which lack statistical robustness of resolution in all three studies.

At the level of the species groups *Teleocichla* is in both *Piálek et al. (2012)* and *Burress et al. (2018a)* nested within *Crenicichla* as the sister group of the Wallacii, Saxatilis and Lugubris groups, in *Burress et al. (2018a)* with all analyses of the ddRAD data using various settings, parameters and matrix sizes giving a 100% bootstrap support for this phylogenetis position. In our mtDNA phylogeny in this study the position of *Teleocichla* is equivocal, either nested within *Crenicichla* (MP and ML analysis; Figs. S1, S2A and S2B), or as a sister group of *Crenicichla* (BI analysis; Fig. 2), in both cases without statistical support. The uncertainty of *Teleocichla* position in mitochondrial markers is likely due to saturation at this large phylogenetic distance (see *Piálek et al., 2012* for details). Based on available studies (*Piálek et al., 2012*; *Burress et al., 2018a*; this study) *Teleocichla* is thus most probably a species group within *Crenicichla*, the sister group of the Wallacii, Saxatilis and Lugubris groups (including Acutirostris group) (*Burress et al., 2018a*).

The southern groups of *Crenicichla* (Missioneira, Lacustris, Scottii) form a well supported monophyletic clade in our mtDNA analyses (Figs. 2, 5, 6, S1, S2A and S2B) as well as in other studies of *Crenicichla* (*Piálek et al., 2012*; *Burress et al., 2018a*) with the Missioneira being the sister group of the Lacustris. The Scottii group is found strongly nested within the Lacustris group as it is in *Piálek et al. (2012)* and *Burress et al. (2018a)*.

The sucessive sister groups of this southern clade are the Macrophthalma group and the Reticulata group, both relationships being well supported in our mtDNA analyses (Figs. 2, 4–6, S1, S2A, S2B) as they are in the mtDNA + nDNA multilocus study of *Piálek et al. (2012)*, while in the nDNA ddRAD study of *Burress et al. (2018a)*, the Reticulata group is the sister group of the southern clade followed by the Macrophthalma group

(bootstrap supports for this alternative relationship have values of 60%, 79%, 86%, 88% and 99% in the various analyses of the ddRAD data in *Burress et al., 2018a*).

The remaining species groups (Wallacii, Saxatilis, Lugubris and Acutirostris) form the second main clade of *Crenicichla* in the phylogenetic BEAST analysis of our mtDNA data (Figs. 2–4), and with the exclusion of the Wallacii group also in the RAXML analysis (Figs. S1, S2A and S2B). The relationships between the Saxatilis, Lugubris and Acutirostris groups however differ between the BEAST and RAXML analyses and lack statistical support. Both *Piálek et al. (2012)* and *Burress et al. (2018a)* found Lugubris and Acutirostris as the sister group of Saxatilis followed by Wallacii, which is a different relationship from both our mtDNA phylogenies with lack of statistical support at these nodes (Figs. 2–4, S1, S2A and S2B), while the topology in both *Piálek et al. (2012)* and *Burress et al. (2018a)* is strongly supported.

The Acutirostris group (and the Scottii group; see above) is the only species group whose species composition and monophyly is questioned by molecular phylogenetic analyses. In our mtDNA analyses the Acutirostris group appears as a separate clade but without statistical support that includes *C. acutirostris, C. multispinosa, C. ternetzi* and also *C. cincta* (Figs. 2–4, S1, S2A and S2B). In the nDNA ddRAD phylogeny of *Burress et al. (2018a)* the group is not monophyletic but is paraphyletic to the Lugubris group (*C. acutirostris* at the base followed by *C. multispinosa* followed by *C. cincta*) with strong statistical support and hence probably best subsumed into the latter group, as has been proposed in *Stawikowski & Werner (2004)*, where it is treated as a species complex within the Lugubris group.

In summary our analyses together with previous molecular studies of *Crenicichla* support the monophyly of the Lacustris, Lugubris, Macrophthalma, Missioneira, Reticulata, Saxatilis, and Wallacii species groups which form two main clades of *Crenicichla*, the Lacustris-Missioneira-Reticulata-Macrophthalma clade, and the Lugubris-Saxatilis-Wallacii clade, and we subsume the Acutirostris group into the Lugubris group and the Scottii group into the Lacustris group.

## Species diversity

Our results clearly demonstrate that the species diversity in *Crenicichla* is significantly underestimated. While our study included 77 valid species our analyses delimited between 126 and 168 species (Figs. 3–6), at least twice as many. Extrapolation from the ratio of total valid species (105) and the number of valid species included in our study (77) (105/77) and the here delimited species (126–168) suggests the minimum number of species in *Crenicichla* to be 172–229 (Figs 3–6).

While the profesional ichthyological knowledge of the species diversity in the group is very limited as seen from our results, the cichlid-hobby knowledge of the *Crenicichla* fauna is actually much better. This is evident from the correspondence between the here delimited species and the potentially undescribed species identified in the cichlid hobby literature. The cichlid hobby literature identifies at least 85 potentially undescribed species of *Crenicichla* and *Teleocichla* (Appendix 3) and essentially all of those that were included in the present study (50) have been delimited by the molecular analyses

(40–46 depending on the delimitation method; Figs. 3–6; this group of delimited taxa shown in violet). Adding the at least 85 identified potentially undescribed species to the presently valid species (105) still gives an underestimated total of 190 *Crenicichla* plus *Teleocichla* species compared to our extrapolation based on molecular species delimitation which is in the range of 172–229 species. Virtually all of these species unrecognized by the professional literature are found in the Amazon-Orinoco-Guiana core of South American fish diversity (*Albert & Reis, 2011*; *Van der Sleen & Albert, 2018*), where they in the *C. lugubris* and *C. saxatilis* groups actually form the majority of the delimited species diversity (Figs. 3–6; delimited species 31–41, 32–51 *vs.* included valid species 12, 19, respectively). Many of the delimited species in this category are presently subsumed under described species and especially under those with putatively large distribution areas. Our mtDNA phylogeny and species delimitations together with their various degrees of morphological distinctiveness clearly demonstate that these putatively widely distributed species are an oversimplification that does not reflect true diversity.

In the following paragraphs we provide a short review of the putative new species previously proposed in the cichlid-hobby literature and here for the first time delimited and supported in molecular analyses. We provide this review by species groups, starting with the *C. lugubris* and *C. saxatilis* groups, where this hidden diversity actually forms the majority of the species diversity.

### Lugubris group

In the Lugubris group the species *C. lugubris* and *C. johanna* have in the present ichthyological definition very large distribution areas covering virtually the whole of the Amazon/Orinoco/Guiana core. Our analyses demonstrate that these and most other widespread species are an illusion and that they are actually species complexes composed of allopatric/parapatric species that have in most instances been separated for millions of years (Fig. 3). The type of *C. lugubris* is from the Negro/Guianas area, in our analyses clearly delimited as a separate species, and our analyses further confirm the proposed allopatric species *C.* cf. *lugubris* Orinoco (here actually as three delimited species *C.* cf. *lugubris* Caroní, *C.* cf. *lugubris* red, *C.* cf. *lugubris* Caura), *C.* cf. *lugubris* Atabapo-Ventuari, all from the Orinoco basin, and *C.* cf. *lugubris* Amazon, *C.* cf. *lugubris* Tocantins, *C.* cf. *lugubris* Tapajos I, and *C.* cf. *lugubris* Tapajos II, all from the Amazon basin. The divergence within the presently recognized C. *lugubris* sensu *lato* ranges from 0.6 to 7.9 My with a mean of 3.5 My based on our calibration (Fig. 3) demonstrating long isolation between the allopatric species sensu *stricto*. The *C. lugubris* group additionally includes several very clearly distinct species that have been known for a long time and that still await description. All that were included in the present study are delimited by our molecular analyses. These are *C.* sp. Atabapo, *C.* sp. Xingu I, *C.* sp. Xingu II (also known as *C.* cf. *lugubris* Xingu), and *C.* sp. Xingu III (recently described as *C. dandara* Varella & Ito, 2018). Apart from the here sampled proposed allopatric species from within the *C. lugubris* complex our analyses do not include *C.* cf. *lugubris* Uaupés and *C.* sp. Uaupés (recently described as *C. monicae* Kullander & Varella, 2015), which both represent clearly distinct morphologically-diagnosed species.

The type of *C. johanna* is from the western Amazon, more precisely from the Rio Guaporé, and while we were not able to sample *C. johanna* from this particular river basin, our analyses clearly delimited other western Amazonian samples as a distinct species. Our analyses further confirm the proposed allopatric species *C.* cf. *johanna* Orinoco I, *C.* cf. *johanna* Orinoco II, *C.* cf. *johanna* Amazonian Guiana, and *C.* cf. *johanna* Eastern Amazon. We have not sampled *C. johanna* from several other large tributaries that probably also represent distinct allopatric species (Tapajós, Xingu, Tocantins, Guianas, Negro, Branco, Aripuanã-Roosevelt, etc). The divergence within the here analyzed C. *johanna* sensu *lato* ranges from 1.2 to 6.7 My with a mean of 3.5 My based on our calibration (Fig. 3) demonstrating long isolation between the allopatric species sensu *stricto*.

Several other species within the *C. lugubris* group have rather large distribution areas (*e.g. C. cincta*, *C. lenticulata*, *C. marmorata*, *C. strigata*) but most of them we have only been able to sample from single river basins. For *C. lenticulata* our analyses do find hidden diversity with the type of *C. lenticulata* from the Rio Negro in our analyses clearly delimited as a separate species from *C.* cf. *lenticulata* Orinoco. The divergence within the presently recognized C. *lenticulata* sensu *lato* is 1.0 to 1.8 My based on our calibration (Fig. 3). On the other hand in *C. marmorata* which we have sampled from several different basins (*e.g.* Xingu, Arapiuns, Curuá) our delimitation analyses do not converge on the same number of species, but the low genetic distances only appear to suggest the presence of a single species.

### Saxatilis group

The Saxatilis group is the most complicated group both in traditional systematics and in our results of molecular phylogeny and species delimitation. The reasons are strong morphological similarity of the species and short internodes between some of the species (Figs. 4, S2A and S2B). The *Crenicichla saxatilis* group contains a large number of generally weakly diagnosed species and most of the unresolved or unsupported nominal species of *Crenicichla* in our analysis are found in this species group (and in the even younger Missioneira group). Despite the limited morphological variability and the young age of some of the species our delimitation analyses delimit the largest number of species of all *Crenicicichla* groups in this species group (32–51 species). The group is thus very complicated yet clearly contains a large diversity of species. Most of the nominal species have distributions in the Guianas and fall into the informally called saxatilis complex, as opposed to the second main complex, the lepidota complex, whose nominotypical species are from southern tropical South America. Both complexes are only weakly defined and overlap throughout Amazonia, but our mtDNA phylogeny largely separates them as two monophyletic groups. There are additionally largely endemic complexes in the western Amazon and in the Orinoco basin which appear in basal positions of the whole group and of the two main complexes in our mtDNA phylogeny.

Within the smaller Orinocoan complex our analyses delimit all the proposed Inirida species (*C.* sp. Inirida I, *C.* sp. Inirida II, *C.* sp. Inirida III), *C.* sp. Caura and *C.* cf. *alta* as
distinct and distantly related to *C. alta* (nominotypical from Guyana; in our sampling from Venezuelan tributaries of Guyana).

Within the western Amazonian complex our analyses delimit multiple species within the nominal *C. anthurus* (*C. lucius* ?, *C. proteus* ?) and *C. semicincta* (and *C.* cf. *semicincta*), but the species boundaries between these nominotypical species require inclusion of topotypical samples and of denser area sampling given the wide putative distribution of these species throughout the western Amazon.

Within the large lepidota complex our analyses delimit many species in this complex with a high degree of endemism. The terminal clade of the complex is made of three clades of *C. edithae*, two in the Middle Paraná (one also including Pantanal) and a separate clade of *C.* aff. *edithae* Iguazú. The sister-group of this clade is composed of many delimited regional endemics from the south of the eastern Amazon (*C. ploegi*, *C. isbrueckeri*, *C.* cf. *lepidota* Teles Pires, *C.* cf. *lepidota* Juruena, *C.* cf. *lepidota* Xingu) and Upper Paraná (*C. britskii*), followed by *C.* cf. *lepidota* Guapimirim from the coast.

The large saxatilis complex has most of its named species in the Guianas and the eastern Amazon but our analyses fail to delimit many of the named species (*C. albopunctata*, *C. coppenamensis*, *C. brasiliensis*, *C. nickeriensis*, *C. inpa*) due to low divergences and lack of reciprocal monophyly. On the other hand our GMYC and bPTP analyses delimit a large number of separate species in the south of the eastern Amazon and its tributaries (in French Guiana, Xingu, Tapajos, Tocantins), but the mPTP analysis delimits them as a single unit (shown as *C.* aff. *saxatilis* or *C.* aff. *inpa* in Fig. 4).

### *Wallacii group*

In the Wallacii group our analyses find hidden diversity in the nominal species *C. regani* and C. *notophthalmus* (Fig. 3). *Crenicichla regani* has in the present ichthyological definition a large distribution area in the eastern Amazon. The type of *C. regani* is from the Rio Trombetas (*i.e.* Amazonian Guiana) and is in our analyses clearly delimited as a separate species (actually two in GMYC and pPTP analyses; Fig. 3) distinct from the allopatric populations of *C.* cf. *regani* Amazon and *C.* cf. *regani* Tocantins. The nominal *C. regani* occurs throughout all of eastern Amazon in many of its tributaries and based on color patterns probably represents several allopatric species. One of the forms found in the Xingu treated previously provisionally as either *C.* cf. *regani* Xingu or *C.* cf. *urosema* Xingu was recently described as *C. anamiri* Ito & Py-Daniel, 2015, but was not sampled in our analyses. The divergence within the here analyzed *C. regani* sensu *lato* ranges from 0.8 to 4.2 My with *C. wallacii* nested within with a 3.2 Ma divergence based on our calibration (Fig. 3) demonstrating long isolation between the allopatric species sensu *stricto*. The type of *C. notophthalmus* is from the Rio Negro by Manaus and is in our analyses clearly delimited as a distinct species. Undescribed populations from the Orinoco basin, referred to as either *C.* cf. *notophthalmus* or *C.* sp. Orinoco in the cichlid hobby literature are in our analyses composed of highly divergent clades, distinct from the nominal C. *notophthalmus* and delimited as at least three distinct allopatric species *C.* cf. *notophthalmus* Orinoco I (Ventuari), *C.* cf. *notophthalmus* Orinoco II (Inirida/Colombia), and *C.* cf. *notophthalmus* Caroní. The divergence between the nominotypical

*C. notophthalmus* and *C.* cf. *notophthalmus* in the Orinoco basin is fully 10.0 Ma and within the here analyzed *C.* cf. *notophthalmus* in the Orinoco basin ranges between 3.4 to 5.9 My based on our calibration (Fig. 3) demonstrating long isolation between the allopatric species sensu *stricto*.

### Reticulata group

In the Reticulata group our analyses delimit all included valid species (Fig. 4). The species *C. reticulata* and *C. cyanonotus* have in the present ichthyological definition very large distribution areas, the first covering virtually the whole of the west-east Amazon including Rio Branco and Amazonian Guiana, and the second the SW Amazon including upper Madeira. Our geographical sampling of both species in this study is still limited, but our analyses demonstrate that nominotypical *C. reticulata* from the Rio Negro (Rio Branco in our sampling) is delimited as a distinct species from the W-E Amazonian *C.* cf. *reticulata* with a divergence of 1.7 My. We have only been able to sample *C.* cf. *cyanonotus* from the upper Madeira and not the nominotypical *C. cyanonotus* from the Western Amazon sensu *stricto* in this study so the taxonomic composition of this species remains to be verified. Our analyses also find *C. jegui* and *C.* cf. *jegui* as distinct, non-sister species with a divergence of 2.8 My.

### Lacustris and missioneira groups

The Lacustris and Missioneira groups are sister-groups and are endemic to southern (sub) tropical South America. In our sampling this monophyletic group includes the largest number of described species (30 including two presently treated as synonyms) and also the densest geographical sampling (Figs. 5, 6). Our analyses delimit in these two species groups 31 to 38 species (25 to 28 and 6 to 10, respectively) and thus our delimitation analyses in this species group are in the best agreement with present classification among all *Crenicichla* species groups. The reason for this good correspondence is that these two groups have recently been revised in several studies (*Kullander & Santos de Lucena, 2006*; *Casciotta, Almirón & Gómez, 2006*; *Lucena, 2007*; *Casciotta & Almirón, 2008*; *Kullander, 2009*; *Piálek et al., 2010*; *Casciotta et al., 2010*; *Kullander & Lucena, 2013*; *Casciotta et al., 2013*; *Mattos et al., 2014*; *Piálek et al., 2015*; *Říčan, Almirón & Casciotta, 2017*; *Piálek et al., 2019a*, *2019b*). These two groups still include many putative undescribed species, only some of which have been included in the present study (see references above). The only notable difference from the generally good correspondence between morphological and molecular delimitations on this group is in the Missioneira group from the Uruguay river basin where five morphologically very distinct sympatric species (*C. missioneira*, *C. minuano*, *C. tendybaguassu*, *C. hadrostigma*, *C. celidochilus*) are not monophyletic in our molecular analyses and nor are they monophyletic in the genomic study of *Burress et al. (2018b)*, probably because of their very young divergence (Fig. 6). The second part of limited molecular delimitation is in the clade including the Iguazú species (*C. iguassuensis*, *C. tesay*, *C. tuca*, *C. tapii*) and most Middle Paraná species from the *C. lacustris* group. This clade demonstrates strong mito-nuclear incongruence (*Piálek et al., 2019a*, *2019b*) which indicates that most of the Middle Paraná species

(all except *C. ypo* and *C. hu*) have had their mtDNA swept by Iguazú species (*Piálek et al., 2019a*, *2019b*) which accounts for the lower delimitation success in mtDNA in this clade.

In summary, our analyses delimit ten previously synonymized species as genetically distinct (shown in orange in Figs. 3–6) and at least 39 (*i.e.*, those where all three delimitation analyses agree; 40–46 in individual delimitation analyses) previously proposed putative new species as genetically distinct (shown in violet in Figs. 3–6), most of them in the Lugubris and Saxatilis groups (*Teleocichla* 2, Wallacii 5, Lugubris 16, Saxatilis 12, Reticulata 2, Lacustris 2, Missioneira 0). Our analyses also identify putative species without any previous reference (shown in red in Figs. 3–6). These total a minimum of 18 (mPTP analysis) while 37 are the shared minimum between the GMYC and bPTP delimitations, and most analyses agree on 25 of these putative species, most of them again in the Lugubris group (*Teleocichla* 1, Wallacii 0, Lugubris 12, Saxatilis 6-8, Reticulata 1, Lacustris 4, Missioneira 1). These delimited units actually equal to one third of the presently known species diversity, a significant portion, but their status is much less clear than for those delimited species that have previously been identified as putative species by cichlid hobbyists. The species level status of these delimited entities remains to be investigated with nDNA data and with careful morphological study of both living and preserved specimens.

## Underestimated Neotropical fish biodiversity

The minimum percentage increase of possibly unrecognised species observed here in *Crenicichla* (65–121%) significantly surpasses the conclusions of *Reis et al. (2016)*, who estimated that 34–42% of Neotropical freshwater fishes remain undescribed. The percentage increase identified here varies widely between *Crenicichla* species group from 0% in the Missioneira and Macrophthalma groups, through 25–40% in the Lacustris group, 50–87% in the Reticulata group, 60–80% in *Teleocichla*, 68–168% in the Saxatilis group, 125–200% in the Wallacii group, and 158–241% in the Lugubris group (Fig. 1). The numbers clearly indicate that the supposedly widely distributed species in the Reticulata, Wallacii, Saxatilis and especially Lugubris group have no foundation in the studied molecular markers. In agreement with *Reis et al. (2016)* the localization of the unrecognized diversity in *Crenicichla* is also in the Amazon basin, especially in its eastern portion and in the remaining two areas of the AOG core (Greater Amazonia) of South American ichthyological diversity.

The general explanations for the unrecognised diversity of Neotropical fishes stem largely from (1) lack of systematic sampling throughout the distibution of the given group and lack of sampling in areas of difficult access, (2) widespread taxa or heterogeneous taxa with insufficient or overwhelming amounts of museum material or improperly preserved material and information, and/or (3) cryptic or pseudocryptic (morphological differences apparent but overlooked) diversity in widespread species (*Reis et al., 2016*). Genetic data are an important instrument in uncovering cases of the latter. In *Crenicichla* the explanation for the underestimated diversity is the same with two additions, namely (1) limitations in available diagnostic characters due to predominant study of preserved specimens which have lost the diagnostic coloration pattern characters

only present in live fishes (this is a specific cichlid problem among all other Neotropical fish groups), and (2) lack of acknowledgement in the professional ichthyological literature of the effort brought by the hobbyists, in this case the cichlid amateurs. As our study demonstrates, the majority of the genetically delineated putatively new species are not unexpected units (of these there are conservatively 25 in our study), but rather previously proposed potentially undescribed species by the cichlid hobbyists (of these there are conservatively 39 in our study).

Therefore, we stress the conclusions of *Reis (2013)*, who demonstrated that the primary instruments for recognition, understanding, and conservation of Amazon basin fishes is increasing expertise in fish taxonomy and systematics and we additionally clearly demonstrate that DNA study and other genetic tools are powerful complementary methods for uncovering fish diversity and highlighting groups in need of taxonomic revisions.

Despite our wide ranging specimen sampling and potential for understanding of *Crenicichla* diversity our study has one limiting factor and that is the use of a single uniparentally inherited molecular marker (mtDNA cytb and ND2 loci). The potential limitations for species delimitation using mtDNA have been discussed in literature (*e.g.*, *Rannala & Zang, 2003*; *Yang & Rannala, 2010*; *Dupuis, Roe & Sperling, 2012*; *Fujita et al., 2012*). The use of additional nuclear markers is generally recommended as the use of these unlinked markers has the potential to improve the accuracy of phylogenetic reconstructions and species delimitation. A study doing just that using the same specimen sampling and the nDNA reduced genome representation ddRAD method is currently in preparation and shows results that are highly congruent with the mtDNA markers employed here.

In spite of having used only mtDNA loci, the majority of molecularly delimited putative species are actually not previously unrecognized but previously synonymized or already postulated species based on coloration patterns, morphological characters and inferred endemism due to potential distribution barriers. All these additional data support and reinforce the inference based on the mitochondrial markers.

## CONCLUSIONS

We find a significantly underestimated species diversity in the largest Neotropical cichlid genus *Crenicichla*. We found between 126 and 168 species-like clusters, *i.e.*, an average increase of species diversity of 65–121% and we found a high degree of congruence between clusters derived from different methods of species discovery. In spite of having used only mtDNA loci, the majority of molecularly delimited putative species are actually not previously unrecognized but previously synonymized or already postulated putative species based on coloration patterns, morphological characters and inferred endemism due to potential distribution barriers. All these additional data support and reinforce the inference based on the mitochondrial markers. Most of the newly delimited putative species are from the Amazon-Orinoco-Guiana (AOG) core area (Greater Amazonia) of the Neotropical region, especially from the Brazilian and Guiana shield areas

of which the former is under the largest threat and largest degree of environmental degradation of all the Amazon.

## ACKNOWLEDGEMENTS

We thank Hernán Ortega (National University of San Marcos, Lima, Peru), Javier A. Maldonado Ocampo (Pontificia Universidad Javeriana, Bogotá, Colombia), Cecilia Rodríguez-Haro (Universidad Regional Amazónica IKIAM, Tena, Ecuador), and Nadacion FUDECI (Venezuela) for supporting our research in their respective countries, and to numerous cichlid hobby friends and research institutions (ANSP, Academy of Natural Sciences of Philadelphia, Pennsylvania, USA; INPA, Instituto Nacional de Pesquisas da Amazônia, Brasil) who helped us obtain a significant portion of the material used in this study.

### Funding

Financial support was provided by the Czech Science Foundation (GACR) grant number 19-20012S and by Comisión de Investigaciones Científicas de la provincia de Buenos Aires (CIC), Facultad de Ciencias Naturales y Museo (UNLP) and Administración de Parques Nacionales. Computational resources were supplied by the Cipres server (https://www.phylo.org/) and the project "e-Infrastruktura CZ" (e-INFRA LM2018140) supported by the Ministry of Education, Youth and Sports of the Czech Republic. The funders had no role in study design, data collection and analysis, decision to publish, or preparation of the manuscript.

### Grant Disclosures

The following grant information was disclosed by the authors:
The Czech Science Foundation (GACR): 19-20012S.
Comisión de Investigaciones Científicas de la provincia de Buenos Aires (CIC).
Facultad de Ciencias Naturales y Museo (UNLP).
Administración de Parques Nacionales.
Cipres server (https://www.phylo.org/) and the project "e-Infrastruktura CZ": e-INFRA LM2018140.
Ministry of Education, Youth and Sports of the Czech Republic.

### Competing Interests

The authors declare that they have no competing interests.

### Author Contributions

- Oldřich Říčan conceived and designed the experiments, performed the experiments, analyzed the data, prepared figures and/or tables, authored or reviewed drafts of the paper, and approved the final draft.
- Klára Dragová performed the experiments, analyzed the data, authored or reviewed drafts of the paper, and approved the final draft.

- Adriana Almirón performed the experiments, analyzed the data, authored or reviewed drafts of the paper, and approved the final draft.
- Jorge Casciotta performed the experiments, analyzed the data, authored or reviewed drafts of the paper, and approved the final draft.
- Jens Gottwald conceived and designed the experiments, performed the experiments, analyzed the data, authored or reviewed drafts of the paper, and approved the final draft.
- Lubomír Piálek conceived and designed the experiments, performed the experiments, analyzed the data, authored or reviewed drafts of the paper, and approved the final draft.

### Animal Ethics

The following information was supplied relating to ethical approvals (*i.e.*, approving body and any reference numbers):

No experiments were conducted on living animals. Study of the animals was approved by the Control Commission for Ethical Treatment of Animals, University of South Bohemia, Faculty of Science, Czech Republic (17864/2005-30/300).

### Field Study Permissions

The following information was supplied relating to field study approvals (*i.e.*, approving body and any reference numbers):

Our collecting was supported by the Administración de Parques Nacionales and Ministerio de Ecología y Recursos Naturales Renovables (permits nos. NEA328 rnv3 and Res: 509/07, respectively).

### DNA Deposition

The following information was supplied regarding the deposition of DNA sequences:

All newly generated sequences (558) are available in GenBank: MW554711–MW584412 (Table S1).

### Data Availability

Phylogenetic matrix of DNA sequences are available in the Supplemental Files.

### Supplemental Information

Supplemental information for this article can be found online at http://dx.doi.org/10.7717/peerj.12283#supplemental-information.

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
