# Peer review of "MtDNA species-level phylogeny and delimitation support significantly underestimated diversity and endemism in the largest Neotropical cichlid genus (Cichlidae: Crenicichla)"

_PeerJ, doi:10.7717/peerj.12283_

## Round 0.1 · original submission · Major Revisions

This is an interesting paper suitable for publishing in PeerJ. The strength of the paper is the large amount of data gathered, however, there are some concerns to deal with before it can be accepted for publication. Please pay close attention to the comments from the three reviewers and modify the paper as you consider adequate.

Reviewer 1 ·

Basic reporting

In my opinion, the writing of the paper is not clear enough and some ideas are repetitive along the manuscript. For instance, a sentence referring to the sample size of the study (lns 33-36): "mtDNA dataset comprising 681 specimens (including Teleocichla, a putative ingroup of Crenicichla) and 77 out of 105 presently recognized valid species (plus six out of 36 nominal synonyms plus over 50 putatively new species)", is repeated 10 times along the manuscript. In the introduction section, the fact that Crenicichla is the largest genus of Neotropical cichlids is mentioned three times. Another example, only in the introduction, the name Crenicichla appears 43 times!. I understand that this is the subject of the study, but that make writing too repetitive and difficult to follow.

Even though English is not my first language, I found several places along the manuscript very difficult to follow, so I would recommend that a native English speaker revise the manuscript and determine all the aspects related to writing.

Regarding the information in the introduction section, I suggest to delete some parts. The list of species included in each species group within Crenicichla is unnecessary and can be referred to an Appendix solely. The same is true for the list of nine species of Teleocichla. Sometimes the authors refer to the genus Crenicichla as the largest in South America, and sometimes to the richest.

My suggestion is to re-structure the entire paper and try to make it more clear, and I also suggest to have an English speaking person editing the paper.

Experimental design

In my opinion the research question is well-defined. The authors try to understand the species diversity in a highly diverse group of freshwater fishes. Their results show that the diversity is underestimated. Again, the problem is the writing and how ideas are expressed repeatedly. For example, in lns 230-233 the authors outline the sampling, but in the preceding paragraph (lns 218-223) it was already mentioned. No methods of euthanasia are referred, and this is important since they include 681 specimens. The authors refer to the fact that collecting was supported by the Administración de Parques Nacionales and Ministerio de Ecología y Recursos Naturales Renovables, is that in Argentina?, and if so, why about all the other countries?. It is also mentioned that specimens were collected through a network of cichlid hobbyists who kept animals during 10 years. Did they have collecting permits to sample in South America?

Analyses regarding DNA sequencing and phylogenetics are ok. However, they are not consistent. It is mentioned that dataset was analyzed through Bayesian inference, maximum lilkelihood and maximum parsimony. Figures only depict the results of BI and in the Appendix, the ML analysis performed in RAXML. The single-locus species delimitation methods are described although there is no mention about the differences between mPTP and bPTP

Validity of the findings

The results obtained by the authors are meaningful. The problem, again, is the way the results are written. It is confusing the way ideas are presented. For example, in lns 377-380 authors said: "Since we included only 77 out of 105 presently recognized valid species and did not include a completely exhaustive areal sampling the minimum number of species in Crenicichla is more likely higher, at least 174 (and up to 232 based on the present sampling; Fig. 3).". Maybe I am wrong, but the sentence is confusing. The same happens with the entire paragraphs of lines 390-401.

One idea of concern is that authors seem to validate the work done by hobbyists which recognize species of cichlids without a formal diagnosis and description. I understand that the experience of aquarists is big, but we have to be careful about that. I understand the the progress made by professionals is not "as large" as that presented in cichlid hobby literature, but I would warn about giving large support to that literature. Actually, in the discussion, authors provide a review of the putative new species previously proposed in the cichlid-hobby literature and that based in their own results through species delimitation analyses. Do not get me wrong, I am not implying that the experience of hobbyists is useless, but is has to be taken with caution.

I would suggest by starting the results presenting the evidence from the phylogenetic analysis and make it clear if that is a concatenated analysis, or is the analysis of each of the 2 mitochondrial genes separately, and define whether it is the result of ML, BI or MP. Then move to explain the species delimitation methods and only present the overall result trying to avoid the use of so many names while describing. See the last paragraph of results (lns 426-435). In 10 lines, the word group is used 22 times!. That makes reading very hard.

Regarding the phylogenetic results, I would insist in showing if that is the result of a combined analysis of the 2 markers. And also, I wonder why Fig. 2 does not present the outgroups referred in the methodology (Astronotus ocellatus and Satanoperca jurupari)?. Is the use of the "groups" name absolutely neccessary?. I think the use of so many names is what makes reading difficult. One of the results indicate that each group is monophyletic, but what does that mean?, what is the point of having a name for a subgroup if all of the belong to Crenicichla?, and then, if removed authors avoid confusion by talking about groups forming subclades and then grups subsumed to other groups, etc.

Another recommendation I would like to make is to shorten the discussion is the species diversity section. Is way too long. My opinion, to make it more clear is just to present an overall description of the recognized species by the delimitation methods and not talk about each group in particular and then close the paper with the section of underestimated neotropical fish diversity which, to me is very relevant.

Finally, nothing is discussed about the validity of the species delimitation results. There is a vast literature out there on the problems associated with the use of such methods in describing biodiversity, especially in metazoans. Some methods tend to overestimate the species diversity, and some are dependent on sample size. This kind of arguments need to be discussed at least briefly in the paper to present a string case of why the methods of choice recovered the "true" history.

Additional comments

I think the results of the paper clearly indicate what the authors mean to show: a significant underestimated diversity in the most species-rich genus of cichlid in South America. This adds a large amount of information to the study of cichlid evolution and diversification. As they also said, this study shows that "DNA study and other genetic tools are powerful complementary methods for uncovering fish diversity and highlighting groups in need of taxonomic revisions". My majos criticism is that the writing is not good enough to present the large amount of information, and that is why my recommendation is to try to make it more concise, to avoid repetitive sentences, and to try to use the less number of names, so readers can follow the main ideas. So, in my opinion, if authors are willing to re-write the paper , have and English specking person reading and editing, and make a more concise presentation of fact, the paper has the potential to be published.

Reviewer 2 ·

Basic reporting

In the present study, the authors gathered an impressive number of samples of species of Crenicichla and Teleocichla and this is without a doubt the strongest point of the study. On the other hand, the authors used only fragments of 2 mitochondrial gene encoding protein genes in their analysis of species discrimination and date, which represents a very weak point of the study. This scenario (many species and few data) makes the study extremely questionable because much of what is presented and discussed is based on topologies without statistical support. In this way, I suggest that the attempt to date be eliminated since with the amount of available data it is not possible to make a robust phylogeny for the genera and that the methods of discrimination of species be revised in order to avoid speculation as to the formation of groups and relationship issues.

The introduction is well written and I suggest the addition of references in the phrases of the lines 76-79, 103-105, 171-180, and 200-207.

Experimental design

In the Line 103 is cited 96 valid species, line 219 = 105 sp, line 371 = 96 sp. The authors should decide which number is the true for Crenicichla.

In Material and Methods the authors says that samples were collect under permit of the Administraci.n de Parques Nacionales and Ministerio de Ecologia y Recursos Naturales Renovables (permits nos. NEA328 rnv3 and Res: 509/07) that I suppose is an Argentina authority. How about collections in other countries?

In the paragraph starting in the line 272 authors justify the use of two genes but both these genes are protein coding genes that evolves at more or less the same rate and are not able to solve correctly the relationships among groups that evolved very recently or are very old. When the see the trees presented (figures 2 and 3) it’s possible to corroborate this since most of the nodes do not present statistical support.

Why authors did not use other popular species delimitation method as ABGD?

In the line 317 the authors says “The first 10% of trees from each run were discarded as burn-in.”. This is not correct since it’s important to discard all trees found before the stabilization of the asymptote curve.

Validity of the findings

As described above the reduced amount of data makes this study very questionable considering the low statistical support of the trees. All conclusions based on this kind or result is also questionable.

Additional comments

In general the study is very interesting but the finds should be reanalyzed in order to reach to a more sustainable conclusion.

Reviewer 3 ·

Basic reporting

The English used in the MS is professional, although some parts require to be more concise.
I considered that the figures required to be better formatted.
The final conclusions of the MS required a strong review in terms of their limitations considering the markers used, and also the main goal of the study fails to completely describe the study.

Experimental design

I consider the MS a valuable contribution to PeerJ, particularly considering the extensive sampling effort of the study, together with the relevance to the phylogeography of the group, however, some reviews are required.

Validity of the findings

The MS requires to clearly state their major findings in contrast to previous studies with larger data sets. Moreover, some conclusions require revision particularly considering the limitations of the markers used, and also the main goal of the study fails to completely describe the study.
The final part of the MS is verbose and at some points not very well organized, possibly a figure could help the authors organizing this section.

Additional comments

The most important concern is related to the inconsistencies observed between the BI and RaxML reconstructions between the species groups with the mitochondrial markers used in the study. In this regard, the authors suggested that markers' saturation could be the main reason for these inconsistencies. However, there is no proof of this saturation in the analyses. Since this is a major goal of the study, I would recommend to the authors, to possibly include some nuclear information that could help them to suggest a more robust hypothesis about the relationships between the two species groups.

Moreover, interestingly the authors present in the MS the BI topology, rather than RaxML, being the latter congruent with the genomic data, thus, recovering Teleocichla nested within the Crenicichla.

Moreover, there is no sense that the authors only described as a major goal of the study the relationship between Crenicichla and Teleocichla, instead of testing the phylogenetic relationships between the major species groups described in the manuscript. I consider this a major point to better describe the main goal of the study.

At some parts of the MS it is difficult to understand the major differences, or systematic advances of the present in contrast to the previously described studies: Piálek et al. 2012 and Burres et al, 2018. Thus, I strongly recommend to the authors to clearly specify in which way this MS corresponds to an improvement of the previous phylogenetic analyses published in the group, particularly considering that some of the major differences could be due to the markers resolution differences, this is particularly relevant in the relationships between Saxatilis, Lugubris, and Acutirostris.

Moreover, the reciprocal monophyly of some groups as Acutirostris group clearly exemplifies the mitochondrial markers resolution limitations, thus, I strongly recommend to the authors, in order to give stronger support to their conclusions, and advance in the knowledge of the group, to include the available nuclear information in their analysis, and in the species delimitation. Otherwise, I found that the MS could quickly become obsolete, in terms of species delimitation, recognizing some paraphyletic groups, and questioning the validity of the study.

Finally, the last part of MS with a short review of the putative new species, I found this part particularly controversial, since as the authors describe there are some troubles with the markers’ resolution.
Moreover, at some parts the divergence times or biogeographic barriers are considered as additional criteria in the species recognition, for example:
541 Our analyses demonstrate that these and most other
542 widespread species are an illusion as explained above and that they are actually
543 species complexes composed of allopatric/parapatric species that have in most
544 instances been separated for millions of years.

However, it would be relevant to clearly specify if these species are also supported with previous studies using genomic data. Moreover, a more robust taxonomic proposal should be considering a combination of nuclear and mitochondrial markers, in this regard, I suggest to the authors to better describe the biogeographic patterns observed in their study, and consider restructuring the final part of the MS.

The last part of the MS considering the underestimation of the species diversity, I consider this part also problematic, particularly considering the wide evidence of DNA barcoding overestimation of the diversity (Song, H., Buhay, J. E., Whiting, M. F., & Crandall, K. A. (2008). Many species in one: DNA barcoding overestimates the number of species when nuclear mitochondrial pseudogenes are coamplified. Proceedings of the national academy of sciences, 105(36), 13486-13491). Similarly, to the previous suggestion, I would recommend to the authors to tone down their conclusions in this sense. Even some parts of this section are valuable to discuss (as the extensive sampling of the present study, or cryptic diversity in the group).

Minor comments:
I found the figure captions somehow confusing and difficult to follow. Also in some of the figures, the information is not well organized. For example in the Figure 2 caption, there is a large description of the image, however, there is no description of the different methods used in the species delimitation. In the figure the colors used in the species delimitation are not clear, this is particularly confusing in figure 6 where more than half of the topology present a gray color but four species are mentioned at the right side of the image (i.e. C. missioneria, C. igara, C. hadrostigma, C. celidochilus).

In the L134, the authors present a list of species included in the study I recommend them to include them in a table.
L.161 please review the phrase.

L200-207, in the major goal of the study, are not included the rest of the species groups included in the study (e.g. Wallaci among the others).

In figure 1, the points could be accordingly to the species groups, since apparently there is clear vicariance among the species.

L230, please clearly specify how many individuals were considered for each species recognized, since the delimitation methods are sensitive to the sample size.

L244, please include the correct codes corresponding to the vouchers.
L323, the description of the molecular clock is poorly described, please clearly specify which were the priors used (e.g. mutation rate).

L368, please specify the number of samples per species.

L424, please clearly specify the inconsistencies among the different methods applied.

L451, this phrase requires revision, particularly since the difference in the relationships between Crenicichla and Teleocichla, could be related to the lack of resolution of the markers used, as was mentioned in L455. Thus, as I mentioned before, it could be important considering to include the nuclear data, in order to contrast the mitochondrial information.

---

## Round 0.2 · Minor Revisions

I have read the new version of the manuscript and the comments provided by the reviewer and I agree with after a minor revision this paper can be accepted for publication.

I think the main modification needs to be done in the discussion where some parts look more like results rather than a discussion but other than that I think the authors did an excellent job at improving the manuscript.

This is an important paper based on a very large data set of a very diverse group of organisms and it will set the stage for further work on the group.


Reviewer 3 ·

Basic reporting

I consider that the English used is clear and professional.
The literature is sufficient in the field.
The article requiere a review in the order of some sections (i.e. Methods), and review in the Discussion, which some parts seems more like results than discussion.
The study represent a relevant study due to the extensive sampling and for sure is relevant from a systematic point of view for this highly diverse group.

Experimental design

-The study represents a relevant contribution for the journal.
-Despite to the use of two mitochondrial markers, the study sampling clearly overcomes some limitations, highlighting the necessity of a wider sampling of some particular groups and/ or regions, in order to increase our knowledge on this group.
-I consider that the study is rigorous, but requieres a better description of the collecting permits from Brazil and Guianas samples.
-I consider that the methods a well described.

Validity of the findings

The study is meaningful.

Yes, the data is robust and well analyzed.

Conclusions are well stated.

Additional comments

MtDNA species-level phylogeny and delimitation support significantly underestimated diversity and endemism in the largest Neotropical cichlid genus (Cichlidae: Crenicichla) (#57123)


Authors: Oldřich Říčan, Klára Dragová, Adriana Almirón, Jorge Casciotta, Jens Gottwald, Lubomír Piálek

I consider that the review made by the authors considerably improved the description of the study, which overall describes a greater diversity within the genus Crenicichla than previously reported. On the other hand, the authors make a detailed description of the species groups, with some possible implications for the taxonomy that could be further explored.

However, I consider that the document still has some problems of missing information, or even lack of order in some parts. Particularly in the methods section, I consider that it is important that the authors describe with more seriousness the import-export permits of the species from Brazil and Guianas, because precisely those regions are among the most extensively sampled in the study, and some taxonomic implications are clearly based in the reliability of the locality of origin of the samples (for example L392-401).

On the other hand, it is important that some sections of the methods are organized, in this regard, markers used are described first than the methods used to obtain the DNA.

Finally, in the discussion, some paragraphs are more results than discussion (L597-611).

Minor Comments:
species-like groups, if it is a boundary, species numbers, the other resulting term ambiguous.
L83: and a large leftover.
107: ‘the best review’, this phrase requires a better explanation, since it could be relative to other studies, based on particular criteria, but all aspects, again this phrase is vague.

L136: "True species", this also does not seem exact, since it is not a question of true or false, but complete or exact, check it.

L318 - Specify the distribution used in the node calibration.

Sometimes the complexes are in subtitles and sometimes they are not (L581, L588, L597, etc.)

L625: sensu lato in italics
L627: sensu stricto in italics

---

## Round 0.3 · accepted · Accept

The authors have incorporated or explained otherwise the main comments made by the reviewer.